# Integrative Gene Expression and Metabolic Analysis Tool *IgemRNA*

**DOI:** 10.3390/biom12040586

**Published:** 2022-04-16

**Authors:** Kristina Grausa, Ivars Mozga, Karlis Pleiko, Agris Pentjuss

**Affiliations:** 1Department of Computer Systems, Latvia University of Life Sciences and Technologies, Liela Street 2, LV-3001 Jelgava, Latvia; kristina.grausa@gmail.com (K.G.); ivars.mozga@gmail.com (I.M.); 2Laboratory of Precision and Nanomedicine, Institute of Biomedicine and Translational Medicine, University of Tartu, 50411 Tartu, Estonia; karlis.pleiko@lu.lv; 3Faculty of Medicine, University of Latvia, LV-1586 Riga, Latvia; 4Institute of Microbiology and Biotechnology, University of Latvia, Jelgavas Street 1, LV-1004 Riga, Latvia

**Keywords:** genome-scale metabolic modeling, transcriptomics, software engineering, Cobra Toolbox 3.0, MATLAB, flux balance analysis, flux variability analysis, omics data analysis

## Abstract

Genome-scale metabolic modeling is widely used to study the impact of metabolism on the phenotype of different organisms. While substrate modeling reflects the potential distribution of carbon and other chemical elements within the model, the additional use of omics data, e.g., transcriptome, has implications when researching the genotype–phenotype responses to environmental changes. Several algorithms for transcriptome analysis using genome-scale metabolic modeling have been proposed. Still, they are restricted to specific objectives and conditions and lack flexibility, have software compatibility issues, and require advanced user skills. We classified previously published algorithms, summarized transcriptome pre-processing, integration, and analysis methods, and implemented them in the newly developed transcriptome analysis tool *IgemRNA*, which (1) has a user-friendly graphical interface, (2) tackles compatibility issues by combining previous data input and pre-processing algorithms in MATLAB, and (3) introduces novel algorithms for the automatic comparison of different transcriptome datasets with or without Cobra Toolbox 3.0 optimization algorithms. We used publicly available transcriptome datasets from Saccharomyces cerevisiae BY4741 and H4-S47D strains for validation. We found that *IgemRNA* provides a means for transcriptome and environmental data validation on biochemical network topology since the biomass function varies for different phenotypes. Our tool can detect problematic reaction constraints.

## 1. Introduction

Advancements in genomics in the last decade have led to a rapid increase in published genome-scale sequences of different organisms. Currently, there are over 2.2 × 10^8^ available genome sequence data samples (https://www.ncbi.nlm.nih.gov/genbank/statistics/on (accessed on 6 May 2021). The growing genome data availability and technological advancements in sequencing and mass spectrometry have contributed to increasing multi-omics dataset generation. From a purely mathematical and statistical point of view, multi-omics dataset analysis is still very challenging and lacks proper methods [1,2]. Genome-scale metabolic models are comprehensive collections of known biochemical reactions catalyzed by specific associated proteins coded by genes. They have been successfully used to explain genotype–phenotype relationships for prokaryotic, eukaryotic, unicellular, and multi-tissue organisms. Genome-scale metabolic modeling (GSM) has already proven to be successful, leading to several significant breakthroughs in health and systems medicine fields [3,4,5], biotechnology [6,7,8], and many other life science fields [9,10,11]. GSM is a method for analyzing the phenotype responses of an organism by calculating flux distribution and other parameters [12] to see the carbon distribution potential and environmental perturbation impact on the metabolism [13,14]. Despite the knowledge of genotype in GSM, phenotypic responses are far from being fully understood. With advances in omics technologies, it has become possible to quantitatively monitor gene transcription (high throughput sequencing), protein expression (mass-spectrometry proteomics), and metabolomics data, narrowing this gap between genotype and phenotype [15,16]. In contrast to gene set enrichment analysis, where genes exhibiting similar biological characteristics are sorted and classified in clusters [17], GSM-based integration methods also consider the interconnectivity of a biochemical network, the steady-state assumption, and gene–protein–reaction (GPR) associations. This allows the analysis of transcriptomics datasets to be performed in an interconnected manner on the biochemical network topology [18] using the gene–protein–reaction rule. There are two fundamental approaches to expression data integration in GSM: (1) directly integrating transcriptome data into model flux bounds (DIRECT) or (2) distributing the reactions into different categories based on transcriptome levels (DISTRIBUTE). The first DIRECT approach was developed by Åkesson [19]. They assumed that very low gene expression levels are associated with non-flux reactions. Published later, E-Flux [20] allowed to integrate quantitative transcriptome measurements directly into the model as the maximum possible flux value (flux upper bound). However, the Gene Inactivity Moderated by Metabolism and Expression (GIMME) [21] algorithm was made to distribute reactions into groups of active and inactive by comparing the corresponding enzymatic gene expression levels to a user-specified threshold and minimizing fluxes through inactive reactions.

The tool *IgemRNA* allows to use DIRECT and DISTRIBUTE transcriptomics data integration approaches. *IgemRNA* is an open-access toolbox created for transcriptomics data statistical and biochemical network topology-based analysis. *IgemRNA* was developed in the MATLAB environment to utilize the up-to-date and most commonly distributed GSM tool Cobra Toolbox 3.0 [22] and spreadsheet file (xls, xlsx) capabilities. *IgemRNA* is designed to analyze the quantitative genome-scale transcriptome data measurements like RNA-sequencing [23] and a targeted gene group transcriptome data, for example, gene expression microarray analysis [24]. *IgemRNA* allows to analyze transcriptome data directly or to integrate it into a metabolic model and perform optimization methods, such as FBA [25,26] or FVA [27]. Metabolic models without omics data cannot reveal distinct phenotype properties and can predict only the theoretical carbon and other chemical element distribution within the model. Applying transcriptome data to a metabolic model makes it possible to explain the phenotype properties in different stress and environmental conditions [28]. Although various tools for omics data integration into GSM already exist and have shown to be practical [29], they are not compatible. They use multiple standards and programming environments (Appendix A). These problems have raised the need for a tool that combines most of the previously published basic functionalities, can be used in conjunction with already available GSM tools, and allows to select a variety of data integration, processing, analysis, and storage options in a user-friendly way. *IgemRNA* has been designed to combine several functionalities of previously published tools (Appendix A), e.g., metabolic flux optimizations, transcriptome data initialization, and integration methods. The access is provided via a graphical user interface. Additionally, *IgemRNA* allows the inclusion of growth medium composition data along with the transcriptome data from the same environmental conditions. However, the main novelty is the built-in functions for transcriptome data pre-processing, including gene mapping and thresholding, transcriptome non- and post-GSM-based optimization statistical analysis, and the automatic comparison of data between different phenotypes.

## 2. Materials and Methods

*IgemRNA* is developed in the MATLAB programming environment. All functionality parts of the *IgemRNA* toolbox are open-access and freely available under the MIT License. GUI window is created with MATLAB *dialog()* function, and all user interface controls are added to the window using MATLAB *uicontrol()*. *IgemRNA* has dependencies for the MATLAB-based GSM tool Cobra Toolbox 3.0 [22] functionality and spreadsheet file (xls, xlsx) capabilities to store the results. All *IgemRNA* functionalities are compatible with MATLAB versions not older than 2014 (https://opencobra.github.io/cobratoolbox/latest/installation.html (accessed on 6 May 2021)). *IgemRNA* is available on GitHub (https://github.com/BigDataInSilicoBiologyGroup/IgemRNA (accessed on 11 April 2022)) and is designed for transcriptome data analysis with or without a metabolic model network topology application.

### 2.1. IgemRNA Architecture Description

Cobra Toolbox 3.0 and spreadsheet data files (transcriptomics, metabolic model, optional medium data) in xls or xlsx format are required to access all the functionality of *IgemRNA* (Figure 1). *IgemRNA* uses different implemented methods and saves results in spreadsheet files, depending on the user-selected transcriptome data processing methods and analysis tasks. *IgemRNA* can also use the built-in methods Flux Balance Analysis (FBA) and Flux Variability Analysis (FVA) in Cobra Toolbox 3.0 [30] to perform more complex analysis on metabolic model network topology.

### 2.2. Tools Functionality Description

*IgemRNA* has seven different functional modules (Figure 2). *IgemRNA* modules include access to the user-friendly interface with data input options, data initialization, and pre-processing steps (gene mapping and thresholding) and executes user-selected non- and post-optimization analysis tasks. Post optimization tasks utilize some Cobra Toolbox 3.0 functions (FBA, FVA). *IgemRNA* can be started by opening the MATLAB software, navigating to the directory containing *IgemRNA.m* script, and running it. If any post-optimization tasks have been selected in the graphical interface, *IgemRNA* will launch Cobra Toolbox 3.0. The most time-consuming step is Cobra Toolbox 3.0 initialization with the update function. Thereby, it is set to run without updates by default.

The *IgemRNA* software allows choosing which transcriptome data, medium composition, and metabolic model files to load. Further, desirable GPR mapping and thresholding procedures and optimization and analysis tasks can be selected. When all necessary options have been selected, *IgemRNA* automatically determines which non- or post-optimization functionalities will be executed. An essential part of *IgemRNA* is the graphical user interface (GUI) module (Figure 2A), which collects information about uploaded data files and selections for transcriptome data analysis and optimization tasks and passes these parameters to matching functional modules. The GUI window is opened using the MATLAB *dialog()* function. User interface controls are added with the MATLAB *uicontrol()* function. Depending on the input data files, specific functional modules are called afterward. The optimization and post-optimization tasks modules will not run if only a transcriptome data file is provided since they require a metabolic model (Figure 2).

The input data module accepts transcriptome, metabolic model, and medium composition data from uploaded files (Figure 2B). All input files must meet specified standards and criteria to be recognized by *IgemRNA*. Transcriptome data must be located in a spreadsheet xlsx file, where gene names are under the *GeneId* column, and the measured transcriptome values are under the *Data* column. Gene names must be the same as defined in the metabolic model. Allowed metabolic model data file types are sbml, mat, and xlsx, and the structure should be the same as defined in Cobra Toolbox 3.0 [22]. Using BioPax, GPML, and SBML files older than the 3.0 version is possible by converting them to the newest SBML 3.2 version using the online Systems Biology Format Converter (SBFC https://www.ebi.ac.uk/biomodels/tools/converters/ (accessed on 11 April 2022)) beforehand. Uptake rates for substrates can also be defined by manual input. The medium composition file contains data concerning the medium in which the organism grows, the substrate uptake, and product reaction rates (mmol*g^−1^*h^−1^), and optionally a specific growth rate (h^−1^) (Figure 4B).

The transcriptome pre-processing functionality module (Figure 2C) is responsible for preparing transcriptome and metabolic model data for analysis and optimization tasks. Pre-processing of transcriptome data relies mainly on decision parameters: gene mapping and thresholding approach, and different combinations of these decisions influence data processing results and biological data interpretation.

*IgemRNA* requires a threshold parameter that defines a border between differentially expressed genes. Two types of thresholds exist: local and global. Local thresholds are automatically calculated for each gene individually, given at least two samples of transcriptomics data of the same condition. However, a global threshold is used as a unique parameter that applies to all genes [31]. Different combinations of local and global thresholds lead to different analysis and optimization results. *IgemRNA* allows users to choose between three different thresholding approaches in the data pre-processing step (Table 1) [31]:Global T1 (GT1): is designed to analyze the transcriptome datasets using one global threshold. Example case shows that all transcriptome levels above 130 sequencing reads per gene (for a detailed description, see Section 2.3) are considered expressed, and others are considered suppressed. The global T1 threshold approach can be used for one or several phenotype transcriptome datasets.Local T1 (LT1): is designed to analyze transcriptome datasets having one global threshold value and a local rule. Local thresholds set a strict border for particular genes based on their varying gene expression levels across multiple samples to determine whether a gene is expressed or suppressed in a specific dataset. Local thresholds are only applied to those genes with expression levels above the global threshold since genes below the global threshold are automatically seen as suppressed. The example case shows that all transcriptome levels defined by the global threshold above 130 sequencing reads per gene are considered as possibly expressed, and below 130 sequencing reads per gene are considered suppressed. Local thresholds for specific genes determine expression or suppression for genes with expression levels above the global threshold.Local T2 (LT2) is designed to use two global threshold values: upper and lower thresholds. Transcriptome levels higher than the upper global threshold are considered expressed genes and are active, and transcriptome levels below the lower global threshold are considered inactive genes. All genes with expression levels between the upper and lower global thresholds are considered possibly active. Local rules for these genes are calculated across multiple gene expression datasets and applied to determine their activity levels. The Local T2 thresholding approach can be used if several transcriptome datasets are available. An example of Local T2 shows that all gene expression levels above the upper global threshold of 130 sequencing reads per gene are considered active. Gene expressions lower than the global threshold of 50 sequencing reads per gene are considered suppressed.

Each threshold approach has unique properties and is included in *IgemRNA*. Global threshold value input can be manual or automatic. In a manual input scenario, the user provides an exact expression value, whereas in an automated global threshold input scenario, threshold values are calculated based on a user-provided percentile. Local thresholds are always calculated automatically across multiple gene expression datasets during analysis. The selection of a thresholding approach is required for both post-optimization and non-optimization task cases.

Other important choice parameters are the Gene mapping approach and Constraining options used only for post-optimization task execution. A metabolic model with GPR data is required, and gene mapping is performed using the GPR association. The most straightforward case is one gene encodes one protein that catalyzes one reaction (one gene, one protein, one reaction) in a metabolic model. However, the presence of enzyme complexes (multiple genes, one protein), isozymes (multiple proteins, one function), and promiscuous enzymes (one protein, multiple functions) in GSM makes these associations more complex, and they are defined with Boolean AND/OR rules. For transcriptome data implementation in GSM, several in silico gene mapping functions can be performed (Table 2):

*Minimum (MIN)* AND operands in the GPR association are calculated by taking the lowest gene expression value.*Geometric mean (GM)* [32] AND operands in the GPR association are calculated as the geometric mean of the gene expression values.*Maximum (MAX)* OR operands in the GPR association are calculated by taking the highest gene expression value.*Sum (SUM)* OR operands in the GPR association are calculated as the sum of all the gene expression values.

Cobra Toolbox 3.0 has implemented several previously published transcriptome analysis methods [22]. *IgemRNA* uses 4 methods to implement transcriptomics data on reactions in the metabolic model:

*Only irreversible reactions function.* Enzymatic reactions have three different directions in metabolic models: irreversible, reversible, and backward irreversible. This approach constraints only irreversible and backward irreversible reactions in the respective direction.*All reactions* function constrains all reactions: irreversible and backward irreversible reactions in an oriented direction, but reversible reactions are constrained in both directions.*Growth not affecting gene deletion only* option allows for the deletion of only those genes with expression values below the given threshold and which deletion does not affect growth. Cobra Toolbox 3.0 *singleGeneDeletion* analysis with the FBA method is performed before executing gene deletion for those genes. Only if the returned output *grRatio* by *singleGeneDeletion* function is equal to 1 (meaning that the wild type growth is equal to the deletion strain growth) does the gene get deleted.*Meet minimum growth requirements* option allows constraining only those reactions where the gene mapping end value (which is set as a reaction bound) is not below the minimum growth requirements for that reaction. Minimum growth requirements are obtained by creating another context-specific model where only the gene deletion and medium exchange reaction constraining is applied to calculate the Cobra Toolbox 3.0 FBA (*optimizeCbModel*) minimization of growth.

*Non-optimization tasks module* is the most straightforward transcriptome analysis module, which does not require additional Cobra Toolbox 3.0 functionality or a metabolic model. This module includes three predefined different transcriptome data analysis methods:
*Filter high- and low-expression genes*: this method uses chosen threshold data and sorts genes into high-expression and low-expression datasets.*Filter low-expression genes*: this method uses chosen threshold parameters, filters genes with expression levels below the supplied thresholds, and returns them as non-expressed datasets.*Filter up-/down-regulated genes between phenotypes*: This method uses chosen threshold data and filters up- and down-regulated genes from two or more transcriptome datasets. The gene names must match in all datasets.The resulting data are passed to the Spreadsheet module (Figure 2G).*Cobra Toolbox module* is called before post-optimization tasks to calculate FBA and FVA results using Cobra Toolbox 3.0 functions (Figure 2E). This module requires a metabolic model.

*Post-optimization task module* is the *IgemRNA* advanced transcriptome analysis module that uses metabolic models and analyses them with Cobra Toolbox 3.0. The module applies all transcriptome data pre-processing functions (Figure 2C). Each function then has several options, resulting in a different analytical approach. The novelty of *IgemRNA* compared to other tools (Appendix A) is the post-optimization module that has several functions for analyzing context-specific models:

*Filter non-flux reactions*: this functionality filters out enzymatic reactions that do not carry a flux because the coded gene transcription levels are below the chosen threshold value in the pre-processing module (Figure 2C).*Filter rate-limiting reactions*: This functionality finds maximum reaction rates equal to the calculated GPR value based on gene expression data. The function uses the FVA optimization method to calculate the minimal and maximal rate value for each reaction and then filters reactions with upper bounds of the same value as the FVA maximal results.*Flux shifts between phenotypes*: this function compares minimal and maximal fluxes (calculated by FVA) between different phenotypes or datasets, calculating ratios between them.

All functionalities generate results, which are passed to the *Spreadsheet module*.

*Results Module* saves all non- and post-optimization analysis results in spreadsheet files (xls or xlsx).

IgemRNA is available on GitHub (https://github.com/BigDataInSilicoBiologyGroup/IgemRNA) (accessed on 11 April 2022).

### 2.3. RNA Sequencing Data Analysis

A publicly available and previously published [33] gene expression dataset from *Saccharomyces cerevisiae* BY4741 strain and mutant strain H4-S47D were used for analysis. Reads were aligned using STAR aligner [34] and assigned to genomic features using *featureCounts* [35]. Statistical comparisons between samples were made using *edgeR* [36]. The workflow is publicly available on the Galaxy platform [37,38]: https://usegalaxy.eu/u/karlispleiko/w/rna-seq-kp-fromgeosingle-read (accessed on 11 April 2022).

## 3. Results

### 3.1. The Comparison of Available Transcriptome Data Integration Tools

Before designing and implementing the IgemRNA software, we reviewed existing transcriptome analysis methodologies. We discovered many previously published methods and sorted them according to a variety of parameters. To classify transcriptome analysis methods, we chose to categorize them by the following attributes: method name (1), does the method returns a context-specific model (2), thresholding method (3), gene mapping approach (4), requirements to run (5), if it is actively maintained (6), 3rd party software availability (7), and accessibility to build-in statistical analysis methods (8).

These methods differ in terms of their functionality and data pre-processing approaches. The method proposed by Åkesson [19] assumes that very low gene expression levels are associated with no flux reactions. The GIMME (Gene Inactivity Moderated by Metabolism and Expression) [21] method compares two transcriptome datasets, determines active and inactive genes, and minimizes low-expression reactions, thus keeping objective function above a set value. The iMAT (Integrative Metabolic Analysis Tool) [39] allows the integration of transcriptomic and proteomic data into metabolic models. This method groups reactions into high-, moderate- and low-expression and maximizes the high- while minimizes the low-expression reactions. MADE (Metabolic Adjustment by Differential Expression) [40] uses two or more datasets of gene expression data across multiple conditions, creates a sequence of binary expression states to find statistically significant changes in gene expression measurements, and determines high-/low-expression reactions. The TIGER (Toolbox for Integrating Genome-scale Metabolism, Expression, and Regulation) [41] software platform facilitates the conversion of GPR associations into a mixed-integer linear program (MILP), which is used to constrain a metabolic model.

The resulting model combines GPR associations with a transcriptional regulatory network and can be further specified using gene expression data. E-Flux [20] constraints upper bounds for reactions classified as low-expression based on a given threshold and expression data. PROM (probabilistic regulation of metabolism) [42] is used for integrating genome-scale transcriptional regulatory networks into metabolic networks. The algorithm calculates the probability that a gene is active with respect to its transcription factor as specified by expression data. It then constrains the maximum reaction flux by a factor of this probability. The application of transcriptional regulatory networks and in metabolic modeling has also been outlined by TRFBA (transcriptional regulated flux balance analysis) [43]. This tool uses gene expression data from various perturbations as continuous variables and constrains reaction upper bounds to link transcriptional regulatory and metabolic networks. The INIT (Integrative Network Inference for Tissues) algorithm [44] maximizes reactions based on a qualitative confidence score and minimizes reactions associated with low expression. In addition, this method allows a small net accumulation rate for internal metabolites to prevent the removal of necessary reactions. Lee-12 [45] integrates absolute gene expression data directly into the objective function of a constraint-based model instead of constraining the fluxes. The biological objective function is replaced by a function that minimizes the deviation between gene expression levels and the fluxes. mCADRE (Context-Specificity Assessed by Deterministic Reaction Evaluation) [46] instead introduces a core set of reactions that must be present and active based on gene expression data and non-core reactions determined based on gene expression and connectivity evidence. This method also runs a test to ensure the basic functionality of the generated models and was successfully applied to reconstruct 126 human tissue genome-scale draft models. Fang-12 [47] is used to predict flux distribution for a perturbed state based on the differences in gene expression levels relative to a reference condition with precalculated flux distribution. This method also allows minor variations in biomass composition for the perturbed state.

Similarly, ΔFBA (deltaFBA) [48] also estimates differences between two conditions only by calculating the flux differences between the conditions, and it does not require the specification of an objective. RELATCH (relative change) [49] uses gene expression and fluxomic data from a reference state to estimate metabolic changes in a perturbed state for which there is no expression data available. Flux distribution for the perturbed state is calculated by minimizing the adjustment to the reference state. The TEAM (Temporal Expression-based Analysis of Metabolism) [50] method estimates time-course flux profiles using temporal gene expression patterns by combining dFBA (Dynamic Flux Balance Analysis) and the GIMME algorithm. It calculates the flux distribution at each time point and uses flux sum minimization to find the optimal solution. The GX–FBA (Gene-expression FBA) [51] method that integrates gene expression data into flux balance analysis uses the deviation in gene expression levels between a reference state and a perturbed state to define flux constraints for the perturbed state.

Akkeson, GIMME, E-Flux, and ΔFBA use only one global threshold for gene activity determination. PROM employs a predefined 33rd percentile threshold from the average value. INIT has an optional minimum flux threshold and positive/negative weights for each reaction. TEAM calculates the threshold from the M3D microarray dataset database. iMAT sets two different global thresholds as lower and upper bounds. Meanwhile, MADE and TRFBA do not have thresholding.

All transcriptome analysis and optimization methods require a metabolic model and one or more transcriptome datasets. However, some algorithms require more complex experimental data or software to run. For iMAT and GIMME, a specified objective function is needed, while MADE needs access to a mixed-integer linear program solver (like GUROBI) and more than one transcriptome dataset. Multiple gene expression datasets states are also necessary to perform PROM, whereas TRFBA and ΔFBA E-Flux requires a function to convert gene expression levels into fluxes. PROM, TIGER, and TRFBA are developed to apply not only transcriptome environmental and genetic perturbation datasets, but also transcriptional regulatory networks. RELATCH method requires transcriptome and fluxomics datasets from the same conditions. The more advanced method TEAM uses initial composition data and temporal transcriptome and biomass composition data.

Along with different data integration approaches, an attribute not less important is the software availability and regular maintenance. For example, Lee-12 [45] has claimed its availability in Cobra Toolbox, but it is not found in the latest Cobra Toolbox 3.0 version. TIGER serves as a software platform for three previously published methods: GIMME, iMAT, and MADE. However, unfortunately, the platform is not maintained anymore. To our best knowledge, GIMME, iMAT, INIT, TEAM (only for microarray experiments), mCADRE, TRFBA, and ΔFBA are maintained regularly.

Most of the published methods are designed for particular modeling cases. They are not flexible for other scenarios since they require complex experimental data (fluxomics, multiple transcriptome datasets, or temporal data) or the integration of transcriptional regulatory networks. Later published transcriptome analysis approaches are considered ineligible for unified manuscript classification criteria.

*IgemRNA* is a novel tool that has combined many previously listed pre-processing and different transcriptome analysis methods. It allows the analysis of transcriptomics data together with or separately from metabolic models to find high- and low- expression genes or reactions and compare them with other phenotype datasets. *IgemRNA* has implemented more advanced pre-processing functions by combining several previously listed methods. It includes three thresholding, four gene mapping, and four reaction constraining options.

*IgemRNA* is compatible with MATLAB-based software and optionally uses Cobra Toolbox 3.0 functionality. Compared to previously mentioned methods, *IgemRNA* facilitates multiple thresholding and gene mapping approaches and several constraining options for transcriptome data integration into metabolic models. The additional feature is the possibility for the end-user to select different types of reactions (based on their flux and constraints) to extract from the result context-specific models. More advanced analysis methods in *IgemRNA* also include data comparison between different phenotypes. On top of that, all previously mentioned options can easily be selected via the graphical user interface.

### 3.2. IgemRNA Demonstration

The main novelty within this paper is the *IgemRNA* tool itself, combining multiple critical decisions and options for transcriptomics data analysis and integration in metabolic models. The graphical interface of *IgemRNA* makes data entry and pre-processing (thresholding, gene mapping) easier for the end-user (Figure 3). Another valuable feature of *IgemRNA* is the possibility for the user to choose which data analysis tasks to perform, including both non-optimization and post-optimization tasks.

After running the main *IgemRNA* script *IgemRNA.m*, a dialog box is opened where it is necessary to select the transcriptomics data file (xls or xlsx format) in the file upload section (Figure 3A). The transcriptomics data should be organized into two columns *GeneId* and *Data* (Figure 4B), where *GeneId* corresponds to the genes in the metabolic model if one is supplied. The transcriptomics data file can consist of multiple RNA-seq samples of the same conditions as well as different phenotypes. Therefore, the phenotype and sample name must appear as the sheet name since this name will be used in the result files and for selecting phenotypes for comparison (Figure 3F,G).

Having selected a transcriptomics data file, the non-optimization tasks section (Figure 3F) becomes visible. Having chosen a GSM of Cobra Toolbox 3.0 available formats, the post-optimization tasks section (Figure 3G) becomes visible. Optionally, a medium data file (xls or xlsx format), where all the needed reactions and their upper and lower bounds are listed (Figure 4A), can be supplied in the file upload section (Figure 3A).

The following four dialog sections (Figure 3B–E) contain options for data pre-processing. *The thresholding approach* (Figure 3B) and *global threshold value selection* (Figure 3C) options determine methods for transcriptomics data pre-processing to split genes into high- and low-expression groups. It is possible to choose one of three thresholding approaches, namely Global T1 (GT1), Local T1 (LT1), or Local T2 (LT2) (see 2.2 threshold parameter). Depending on the selected approach, the user is then asked to enter one or two global threshold values, which can be supplied as a percentile or an exact value. *Gene mapping approach* (Figure 3D) and *constraining options* (Figure 3E) determine methods for transcriptomics data integration in a metabolic model. The gene mapping approach specifies what operations to use for mapping gene expression data to gene-protein-reaction (GPR) associations (see 2.2 Gene mapping), whereas constraining options allow to constrain only irreversible or all reactions. Moreover, it is possible to constrain reactions (see 2.2 Constraining options) by deleting only non-essential genes and the satisfy biomass objective function minimal resources value.

*Non-optimization tasks* (Figure 3F) allows to filter high- and/or low-expressed genes, export results in spreadsheet and compare gene expression data between phenotypes.

*Post-optimization tasks* combine transcriptomics data and a metabolic model to extract data on reaction levels. It is possible to filter non-flux and rate-limiting reactions and calculate reaction flux shifts between different phenotypes. An objective function is required to perform optimizations on the metabolic model. Since post-optimization tasks use some of the Cobra Toolbox 3.0 functions, the user is also asked to choose how to start Cobra Toolbox: with or without updates.

To validate *IgemRNA,* we used the *Saccharomyces cerevisiae* genome-scale metabolic model [52] version 8.4.0 [53] and transcriptome datasets [33]. We performed high-throughput genetic screenings that provide a novel global map of the histone residues required for transcriptional reprogramming in response to heat and osmotic stress in steady-state growth conditions. All validation details are found in Appendix A.

To perform test cases provided in this user manual, simply run scripts via MATLAB environment. For example, *TestCase_determineGeneActivity.m* script will run a non-optimization task filter for high- and low-expression genes in Appendix A.

## 4. Discussion and Conclusions

Having summarized several transcriptome data processing tools, we found that many are case-specific and allow users to select only a few possible data pre-processing and analysis functions. Recently proposed tools are not widely used due to the complexity of required input data, such as tools that facilitate time-series or genome-scale metabolomics data that are not widely accessible or use their own unique FBA modified functions or different 3rd party software.

We sorted out methods and algorithms providing flexibility and the possibility of application in a wide range of experimental scenarios and implemented them in the newly developed transcriptome analysis tool, *IgemRNA*. The main contributions of the gene expression and metabolic analysis tool are a user-friendly graphical interface, the tackling of compatibility issues by combining several data pre-processing and integration methods in MATLAB environment, allowing the end-user to select different types of reactions (based on their flux or constraints) to extract and filter from the result models, and the novel algorithms for the automatic comparison of transcriptome data from different phenotypes with or without Cobra Toolbox 3.0 optimization algorithms.

We found that *IgemRNA* provides a means for transcriptome and environment data validation on biochemical network topology since the biomass function varies for different phenotypes. The software can detect problematic reaction constraints to loosen their bounds and achieve a steady state. *IgemRNA,* in contrast to gene set enrichment analysis, additionally validates transcriptomics measurement quality, where minimal metabolic network connectivity and flux requirements must be fulfilled. Otherwise, transcriptome data quality is questionable.

In general, the *IgemRNA* tool combines unique data entry, initialization, optimization, and analysis algorithms that significantly facilitate manual work for the end user with the integration of transcriptomics and environmental data into metabolic models, and thus presents great potential to increase novel scientific discoveries in research concerning different organisms.

## Figures and Tables

**Figure 1 biomolecules-12-00586-f001:**
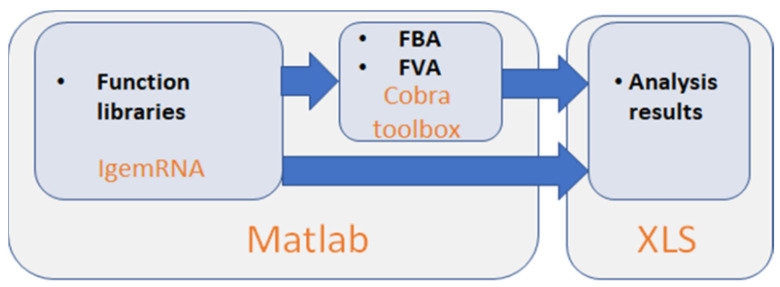
*IgemRNA* toolbox architecture scheme.

**Figure 2 biomolecules-12-00586-f002:**
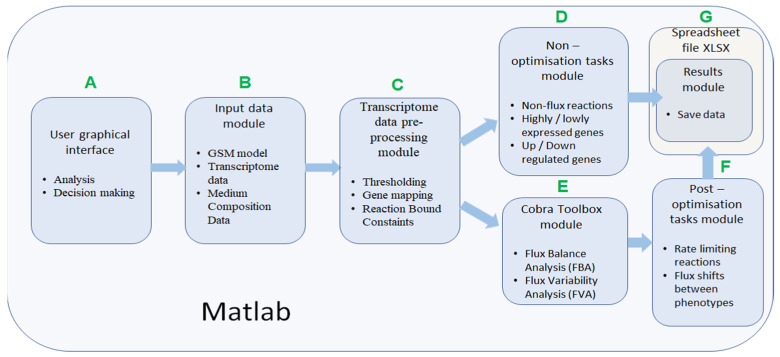
Description of IgemRNA function modules. (**A**) User graphical interface, (**B**) Input data module, (**C**) Transcriptome data pre-processing module, (**D**) Non-optimisation tasks module, (**E**) Cobra Toolbox module, (**F**) Post-optimisation tasks module, (**G**) Spreadsheet file module.

**Figure 3 biomolecules-12-00586-f003:**
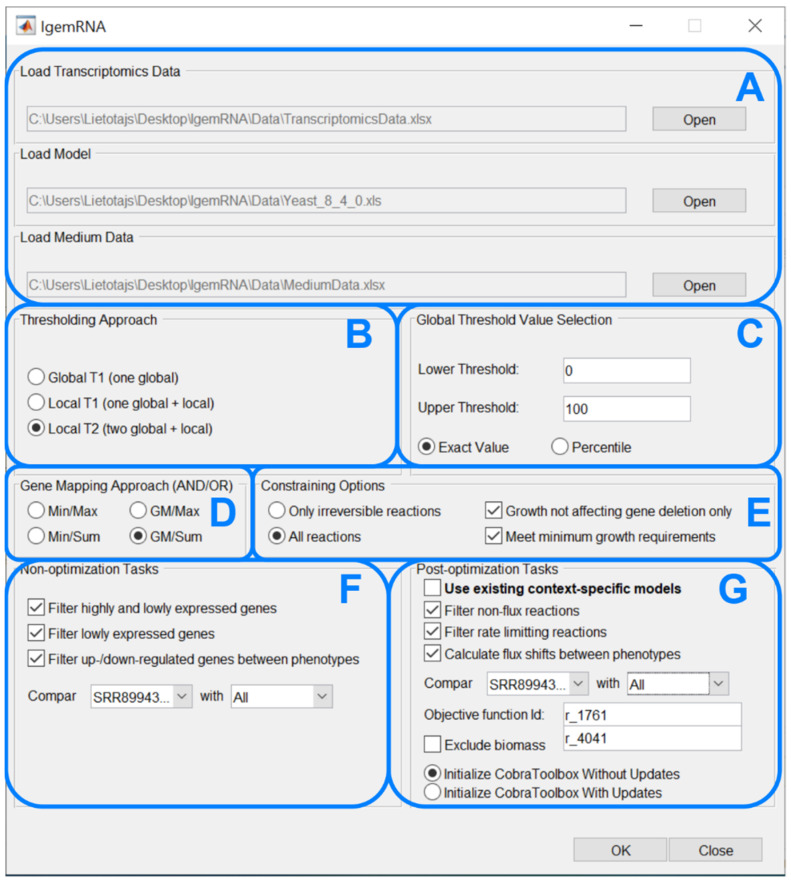
IgemRNA’s graphical interface. (**A**) Input data window, (**B**) Thresholding window, (**C**) Global thresholding value window, (**D**) Gene mapping window, (**E**) Reactions and transcriptomics data mapping window, (**F**) Non—optimization task window, (**G**) Post—optimization window.

**Figure 4 biomolecules-12-00586-f004:**
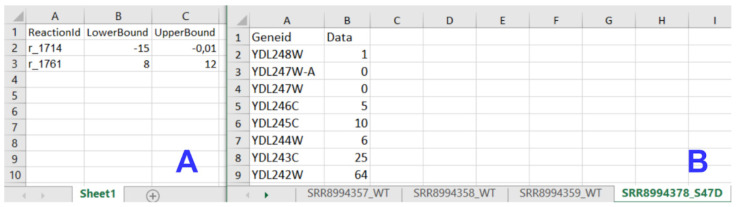
Input data file structure; (**A**) Medium data file structure; (**B**)Transcriptomics data file structure.

**Table 1 biomolecules-12-00586-t001:** Thresholding Options.

Approach	Threshold Input	Visual Representation	Examples	
Global T1	**One global threshold:** exact value input percentile input (25th, 75th, 90th…)	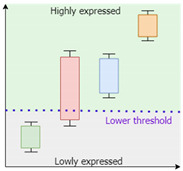	Lower threshold: 130	
			YDL227C	871
			YDL226C	126
			YDL225W	319
			YDL224C	56
			YDL223C	13
			YDL222C	3
			YDL221W	135
Local T1	**One global threshold:** exact value input percentile input (25th, 75th, 90th…) **Multiple local thresholds:** calculated automatically across multiple transcriptomics data samples of the same conditions	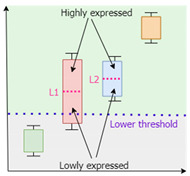	Lower threshold: 130	
			Local for YDL227C: 600	
			Local for YDL225W: 350	
			YDL227C	871
			YDL226C	126
			YDL225W	319
			YDL223C	56
			YDL223C	13
			YDL222C	3
			YDL221W	135
Local T2	***Two global thresholds:*** exact value input percentile input (25th, 75th, 90th …) ***Multiple local thresholds:*** calculated automatically across multiple transcriptomics data samples of the same conditions	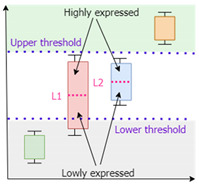	Lower Threshold: 50	
			Upper Threshold: 130	
			Local for YDL224C: 60	
			Local for YDL226C: 70	
			YDL227C	871
			YDL226C	126
			YDL225W	319
			YDL224C	56
			YDL223C	13
			YDL222C	3
			YDL221W	135

**Table 2 biomolecules-12-00586-t002:** Gene Mapping Options.

Requirement	Options
Reaction constraining options	Only irreversible reactions All reactions Non—essential gene deletion only Meet minimum growth requirements
Gene mapping approach	AND/MIN and OR/MAX AND/MIN and OR/SUM AND/geometric mean and OR/MAX AND/geometric mean and OR/SUM

## Data Availability

IgemRNA MATLAB scripts and test cases for transcriptome data integration in genome-scale metabolic models are available at https://github.com/BigDataInSilicoBiologyGroup/IgemRNA (accessed on 11 April 2022).

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
