# Peer review of "Integrative Gene Expression and Metabolic Analysis Tool *IgemRNA"

_biomolecules, 2022, doi:10.3390/biom12040586_

Round 1

Reviewer 1 Report

  • Major Comments
    • Can a matlab live script tutorial with an example application be disseminated to help end users to use the code?
    •  
  • Minor Comments
  • 219 * 106 ? in “In the last decade the amount of available genomic sequences has increased rapidly 33 and counts more than 219 * 106 (https://www.ncbi.nlm.nih.gov/genbank/statistics/ on 34 06.05.2021).”
  • GSM is an overused acronym and in different places means different things. Please write what is meant with full words and not acronyms.
  • Formatting error? “directly into models as the maximum possible flux value (flux upper bound). 62 But GIMME [21] algorithm was made to distribute reactions into groups of highly 63 and lowly expressed by comparing them to a user specified threshold. 64 IgemRNA is an open access toolbox for transcriptome data statistical and biochemical 65 network”
  • “The most time-consuming step is Cobra Toolbox 3.0 initialization with up- 121 date function. IgemRNA has options to run Cobra Toolbox 3.0 with or without an update 122 function.” Why not just set the update option to false by default?
  • Correct this citation format “pre-processing step (Table 1) 165 (https://doi.org/10.1371/journal.pcbi.1007185)”
  • “Example case shows that all transcriptome levels above 130 are considered 168 as expressed and others are considered as suppressed” 130 what? units?
  • Rather than just two approaches of Gene mapping MIN/MAX and MIN/SUM, consider to also add geometric mean for AND. See https://www.nature.com/articles/s41467-019-13818-7

Author Response

Reviewer 1

Comments and Suggestions for Authors

Major Comments

  • Can a matlab live script tutorial with an example application be disseminated to help end users to use the code?

The MATLAB is one of the most popular tools for data analysis in different scientific fields. IgemRNA uses MATLAB environment to gain access to CobraToolbox 3.0 tool functionality to analyse and / or integrate in the metabolic model’s different quantitative transcriptomics and environment data set(s). We developed user manual in Word document where each step by step activities should be done to run specific IgemRNA data integration, optimisation and analysis algorithms. User manual is available in GitHub.

We agree that MATLAB offers live script tutorial to generate user manual with better user graphical interface and allows interactively change IgemRNA inputs and properties.

We will explore the possibilities how to implement IgemRNA functionality and data processing in MATLAB live script pipeline and create interactive user manual in next IgemRNA tool version. Input data is large (BIG) omics data sets and this approach implementation in live script creation will take a long time.

  • Minor Comments
  • 219 * 106 ? in “In the last decade the amount of available genomic sequences has increased rapidly 33 and counts more than 219 * 106 (https://www.ncbi.nlm.nih.gov/genbank/statistics/ on 34 06.05.2021).”

Has been changed according suggestions.

Line 33 - 34

  • GSM is an overused acronym and in different places means different things. Please write what is meant with full words and not acronyms.

Has been changed according suggestions.

Genome-scale metabolic modeling (GSM)

  • Formatting error? “directly into models as the maximum possible flux value (flux upper bound). 62 But GIMME [21] algorithm was made to distribute reactions into groups of highly 63 and lowly expressed by comparing them to a user specified threshold. 64 IgemRNA is an open access toolbox for transcriptome data statistical and biochemical 65 network”

Has been changed according suggestions.

Line 199 - 204

  • “The most time-consuming step is Cobra Toolbox 3.0 initialization with up- 121 date function. IgemRNA has options to run Cobra Toolbox 3.0 with or without an update 122 function.” Why not just set the update option to false by default?

Has been changed according suggestions in IgemRNA scripts and in manuscript text.

Line 503 - 505

  • Correct this citation format “pre-processing step (Table 1) 165 (https://doi.org/10.1371/journal.pcbi.1007185)”

Has been changed according suggestions

Line 600 - 601

  • “Example case shows that all transcriptome levels above 130 are considered 168 as expressed and others are considered as suppressed” 130 what? units?

Has been changed according suggestions

Line 600 – 601

Sequencing reads per gene

  • Rather than just two approaches of Gene mapping MIN/MAX and MIN/SUM, consider to also add geometric mean for AND. See https://www.nature.com/articles/s41467-019-13818-7
  • Has been changed according suggestions in IgemRNA scripts and in manuscript text.
  • Line 647 - 663

Reviewer 2 Report

The manuscript by Grausa et al. reports a very timely development of a user-friendly tool, IgemRNA, which enables to build a context-specific genome-scale metabolic model using integrated transcriptomics data. The manuscript is clear to follow and interesting. Moreover, the authors provided user manual documentation for the IgemRNA tool significantly improving the reproduction procedure of the presented test case study. Overall, the topic of the research appears robust and within the scope of the special issue.

However, despite the Introduction coverage, a scale of the comparative analysis on developed tools, the manuscript does not contain some important references and corresponding tools relevant to the study:

deltaFBA - https://doi.org/10.1371/journal.pcbi.1009589

TRFBA - https://doi.org/10.1093/bioinformatics/btw772

mCADRE - https://doi.org/10.1186/1752-0509-6-153

TIGER - https://doi.org/10.1186/1752-0509-5-147

It will be essential to add these tools to the analysis mentioned in the manuscript.

Furthermore, the authors point out that the developed user-friendly graphical interface of the tool is one of the main contributions. It is absolutely correct, especially for algorithms and methods in the MATLAB environment. However, my major concern is a research outcome that can be obtained by means of the tool compared to others. In this context, I would highly recommend to the authors to conduct the comparative analysis using any classic metabolic model (e.g. smaller size than in the test case study), a unified set of transcriptomics data for all compared tools and demonstrate the reliability of the FBA, FVA results and even show novel biologically meaningful predictions based on the multiple settings for implemented algorithms the IgemRNA tool.

Minor comments:

Line 12 and further: change on “genome-scale…”

Line 68: “The tool…”

Lines 152-153: use the upper case for the power

Line 166: modify the citation according to the journal instructions

Table 2: “OR/MAX” in the first gene mapping approach

Line 251: “…up-…”

Line 305: “…minimizes…”

Line 311: “…data calculates…”

Line 316: “…Lee–12 [42] integrates…”

Line 328: “…calculates the flux…”

Line 347: “…TEAM uses…”

Line 359: “The tool…”

Appendix B.  Figure 17. Specify the caption

Appendix B. italic font style for used names of biological species

Appendix B.  Table 1. Remove point after Figure 3

Author Response

Comments and Suggestions for Authors

The manuscript by Grausa et al. reports a very timely development of a user-friendly tool, IgemRNA, which enables to build a context-specific genome-scale metabolic model using integrated transcriptomics data. The manuscript is clear to follow and interesting. Moreover, the authors provided user manual documentation for the IgemRNA tool significantly improving the reproduction procedure of the presented test case study. Overall, the topic of the research appears robust and within the scope of the special issue.

However, despite the Introduction coverage, a scale of the comparative analysis on developed tools, the manuscript does not contain some important references and corresponding tools relevant to the study:

deltaFBA - https://doi.org/10.1371/journal.pcbi.1009589

TRFBA - https://doi.org/10.1093/bioinformatics/btw772

mCADRE - https://doi.org/10.1186/1752-0509-6-153

TIGER - https://doi.org/10.1186/1752-0509-5-147

It will be essential to add these tools to the analysis mentioned in the manuscript.

Has been implemented according suggestions

Line 816 – 1098

Appendix A

Furthermore, the authors point out that the developed user-friendly graphical interface of the tool is one of the main contributions. It is absolutely correct, especially for algorithms and methods in the MATLAB environment. However, my major concern is a research outcome that can be obtained by means of the tool compared to others.

In this context, I would highly recommend to the authors to conduct the comparative analysis using any classic metabolic model (e.g. smaller size than in the test case study), a unified set of transcriptomics data for all compared tools and demonstrate the reliability of the FBA, FVA results and even show novel biologically meaningful predictions based on the multiple settings for implemented algorithms the IgemRNA tool.

We agree that IgemRNA tool uses several of other previously published transcriptomics integration methods and ideas. However, the goal of this article was to create a new tool combining the best methods which are functionally flexible and can be applied for a variety of modelling scenarios. The goal was not to test and compare the results of previous methods. In manuscript we already concluded, that

“Most of the published methods are designed for very specific modelling cases and are not flexible for other scenarios since they require complex experimental data (fluxomics, multiple transcriptome datasets or temporal data) or the integration of transcriptional regulatory networks. Later published transcriptome analysis approaches are considered ineligible for unified manuscript classification criteria.”

Line 1079 – 1083

Not all previously published methods use the MATLAB environment, have specific  rules of input data properties, uses not only transcriptomics but other omics data measurements for null space analysis.

Thus, previously published tools are designed for specific aims, have different data integration approaches and properties, uses different 3rd party tools and different omics data sets for additional null space constraining.

We concluded that it is impossible to compare all previously published tools and methods, if they are not completely rebuilt for specific programming environment, like Python or MATLAB. We focused only on the most promising and mostly applied data integration, optimisation and analysis methods and algorithms, compiled them and integrated them in the novel IgemRNA tool, which has compatibility with MATLAB environment and most popular metabolic modelling tool Cobra Toolbox 3.0.

All manuscript.

Minor comments:

Line 12 and further: change on “genome-scale…”

Has been changed according suggestions.

Line

Line 68: “The tool…”

Has been changed according suggestions.

Line

Lines 152-153: use the upper case for the power

Has been changed according suggestions.

Line

Line 166: modify the citation according to the journal instructions

Table 2: “OR/MAX” in the first gene mapping approach

Has been changed according suggestions.

Line

Line 251: “…up-…”

Has been changed according suggestions.

Line

Line 305: “…minimizes…”

Has been changed according suggestions.

Line

Line 311: “…data calculates…”

Has been changed according suggestions.

Line

Line 316: “…Lee–12 [42] integrates…”

Has been changed according suggestions.

Line

Line 328: “…calculates the flux…”

Has been changed according suggestions.

Line 347: “…TEAM uses…”

Has been changed according suggestions.

Line

Line 359: “The tool…”

Has been changed according suggestions.

Line

Appendix B.  Figure 17. Specify the caption

Has been changed according suggestions.

Line

Appendix B. italic font style for used names of biological species

Has been changed according suggestions.

Line

Appendix B.  Table 1. Remove point after Figure 3

Has been changed according suggestions.

Line

Round 2

Reviewer 2 Report

The authors have provided a thoroughly revised version of their manuscript. Thank you for your effort.

Author Response

Thanks.